# Association between Mediterranean Diet and Advanced Glycation End Products in University Students: A Cross-Sectional Study

**DOI:** 10.3390/nu16152483

**Published:** 2024-07-31

**Authors:** Nikolina Polić, Viviana Matulić, Tanja Dragun, Helena Matek, Mario Marendić, Ivana Žižić Efendić, Andrea Russo, Ivana Kolčić

**Affiliations:** 1General Hospital Šibenik, Ul. Stjepana Radića 83, 22000 Šibenik, Croatia; nikolina.polic@gmail.com; 2Department of Obstetrics and Gynecology, University Hospital Split, Spinčićeva 1, 21000 Split, Croatia; viviana.hr@gmail.com; 3Department of Physiology, University of Split School of Medicine, Šoltanska 2, 21000 Split, Croatia; tanja.dragun@mefst.hr; 4Family Medicine Practice, Ulica Stjepana Radića 83, 22000 Šibenik, Croatia; helenamatek1@gmail.com; 5University Department of Health Studies, University of Split, Ul. Ruđera Boškovića 35, 21000 Split, Croatia; mmarendic@ozs.unist.hr; 6Health Center Split-Dalmatia County, Kavanjinova 2, 21000 Split, Croatia; ivana.zizic90@gmail.com; 7Faculty of Maritime Studies, University of Split, Ruđera Boškovića 37, 21000 Split, Croatia; arusso@pfst.hr; 8Department of Public Health, University of Split School of Medicine, Šoltanska 2, 21000 Split, Croatia; 9Andrija Stampar Teaching Institute of Public Health, Mirogojska Cesta 16, 10000 Zagreb, Croatia; 10Psychiatric Clinic Sveti Ivan, Jankomir 11, 10090 Zagreb, Croatia

**Keywords:** Mediterranean diet, advanced glycation end products, coffee, physical activity, smoking, health, university students

## Abstract

The aim of this study was to evaluate the association between the Mediterranean diet (MD) and the accumulation of advanced glycation end products (AGEs) measured by skin autofluorescence. This cross-sectional study included 1016 healthy students from the University of Split, Croatia. Participants completed a self-administered questionnaire. Adherence to the MD was assessed using the Mediterranean Diet Serving Score (MDSS), and tissue AGEs accumulation was measured using the AGE Reader mu (DiagnOptics). Multivariate linear regression was used in the analysis. Students’ age and female gender were associated with higher levels of AGEs, which was likewise found for greater coffee intake, adequate olive oil consumption, smoking, and lower levels of physical activity. Higher consummation of vegetables and eating breakfast regularly were associated with lower AGEs levels. The overall MD adherence was not associated with AGEs, possibly due to very low overall compliance to the MD principles among students (8.3% in women and 3.8% in men). Health perception was positively associated with the MD and nonsmoking and negatively with the perceived stress level, while AGEs did not show significant association with self-rated students’ health. These results indicate that various lifestyle habits are associated with AGEs accumulation even in young and generally healthy people. Hence, health promotion and preventive measures are necessary from an early age.

## 1. Introduction

Advanced glycation end products (AGEs) have an important role in the pathogenesis of many chronic noninfectious diseases [1]. AGEs are the product of a nonenzymatic reaction known as the Millard reaction, in which the carbonyl component of a sugar reacts with a free amino group of an amino acid [2]. They can be formed either endogenously or ingested through diet and tobacco exposure, which represent the main exogenous sources of AGEs [3]. Higher cooking temperature and duration, water activity, pH, storage time, and food compounds all have a significant effect on AGEs formation [4,5]. For instance, foods rich in fat and protein; baked, grilled, and frayed foods; and animal products are sources of higher dietary AGEs [6].

AGEs are a heterogenous group of molecules, which contributes to challenges for their quantification in blood, urine, tissues, and cells, while some of them have the property of fluorescence [3]. AGEs are recognized by several receptors, and the signaling pathway for the advanced glycosylation end product-specific receptor (RAGE) is the one most widely studied [3,7]. AGEs binding to the RAGE receptor activate intracellular transcription factors, resulting in the production of proinflammatory cytokines, oxidative stress, and inflammation [8]. Indeed, AGEs have been implicated in the pathophysiology of noncommunicable diseases, especially those associated with oxidative stress and elevated levels of inflammation, such as cardiovascular diseases, obesity, metabolic syndrome, arthritis, autoimmune diseases, chronic renal failure, Alzheimer’s disease, and cancer, as well as with the normal aging process [1,9,10]. AGEs also have an adverse effect on complications of diabetes [10,11], heart failure and systolic cardiac function [12], and possibly on chronic liver disease and hepatocellular carcinoma [8], as well as a contributing role in the pathology of multiple sclerosis [13].

Due to these findings, dietary sources of AGEs are very important, especially because they are regarded as the most important source of AGEs [5,14]. In particular, highly processed foods are characterized with an excessive amount of exogenous AGEs due to their production process using the dry heat technology, as well as their sugar and fat content [6]. On the other hand, the lowest levels of exogenous AGEs are found in vegetables and fruits, which have antiglycation properties, along with herbs and spices [6]. One dietary pattern that is particularly rich in these foodstuffs is the Mediterranean diet (MD). In short, the MD is mostly a plant-based diet that is low in saturated fats and rich in antioxidants, fibers, phytosterols, probiotics, and monosaturated fatty acids, with an adequate omega 6 to omega 3 fatty acid ratio [15]. According to the modern Mediterranean pyramid, each main meal should include a combination of whole grain cereals, vegetables, fruits, and olive oil, along with a daily intake of nuts and fermented dairy products [16]. As protein sources, the preference is given to legumes, along with the intake of fish, eggs, and white meat, which are recommended a couple of times per week [16,17].

Interestingly, the MD was shown to reduce both the serum concentration of AGEs and the expression of the RAGE receptor, and to increase the expression of the AGEs receptor 1 (AGER1), when compared with the consumption of a Western diet [18]. Adherence to the MD lowers dietary AGEs and moves the oxidative balance score to the antioxidant direction, and thereby lowers inflammation [19]. Furthermore, the MD was shown to be associated with a lower incidence of chronic diseases and lower physical impairment in old age, as well as with the reduction in total and cause-specific mortality [20,21]. Specifically, greater adherence to the MD was associated with a lower incidence of coronary heart disease and ischemic stroke [22], lower cardiovascular mortality [23,24], lower rates of both cancer incidence and cancer mortality [24,25], lower incidences of Parkinson’s and Alzheimer’s disease [24], lower incidences of cognitive decline, dementia, and unipolar depression, and a reduced risk of chronic obstructive pulmonary disease [21]. The benefits of the MD extend toward its positive effect on obesity, metabolic syndrome, and diabetes [21], as well as on osteoporosis, sarcopenia, and sexual performance in the elderly [23].

Additionally, the Mediterranean lifestyle is more than just nutrition; it is also promoting affordable, traditional, eco-friendly, and local foods while preserving local culinary and cultural heritage [17]. Another daily habit that is deeply incorporated into the Mediterranean lifestyle is physical activity. Studies have shown a positive impact of moderate physical activity on reducing the risk and improving the prognosis of chronic diseases, such as diabetes mellitus and cardiovascular diseases [26]. On the other hand, lack of physical activity has been associated with increased oxidative stress, which contributes to endothelial dysfunction and atherosclerosis [27]. Physical activity is associated with the reduction of blood pressure, abdominal fat, and improvement in serum lipids levels [28]. Until now, several studies have found a positive effect of physical activity on lowering AGEs in healthy individuals, as well as those already diagnosed with chronic diseases [29,30,31,32,33,34,35].

Besides diet and physical activity, previous studies have identified the effects of several other lifestyle-related factors on the level of AGEs. These include the positive association between AGEs and age, coffee consumption [36,37,38], and cigarette smoking [37,38,39]. Smoking has a significant effect on the accumulation of AGEs, which is affected by the pack-years and the number of hours being exposed to secondhand smoking, indicating a dose-dependent effect [37,39]. It has also been found that if a person quits smoking, the AGEs levels will gradually decrease over time [39,40].

Most of the studies published so far have investigated the association between AGEs and different health-related risk factors and outcomes in middle-aged and elderly subgroups of the population. Additionally, a noninvasive tool for the assessment of AGEs in tissue has been frequently used (the AGE Reader mu of DiagnOptics Groningen, The Netherlands), and it has been validated and shown to be useful in the area of cardiovascular risk assessment. However, this device has not been extensively used in nutrition studies investigating the association between AGEs and the Mediterranean diet or other dietary factors. Hence, there is a knowledge gap among younger individuals who are still free from chronic diseases, especially regarding the relevance of AGEs for their health, as well as the interplay of different factors associated with AGEs levels, such as nutrition and other lifestyle habits. The aim of this study was to investigate the association between AGEs accumulation in the skin and the MD pattern, as well as other lifestyle characteristics in the population of young and healthy university students from Croatia.

## 2. Materials and Methods

This cross-sectional study was performed from December 2018 to May 2019 among Croatian university students as a part of the HOLISTic study [41]. We included three faculties: the University of Split School of Medicine, the University Department of Health Studies (nursing and midwifery students), and the University of Split Faculty of Maritime Studies. The study was approved by the Ethical Committee of the University of Split School of Medicine, and the Ethical Committee of the University Department of Health Studies, while the Faculty of Maritime Studies confirmed those two previously received approvals and rendered them valid for data collection among their students.

The researchers enrolled students during their regular and obligatory courses. Firstly, we obtained consent from both the professors and the students for participation, informing them about the procedures and the purpose of the study. After the introduction, students were asked to answer the self-administered questionnaire anonymously and to undertake AGEs measurement. Hence, all involved participants gave their verbal consent for inclusion in the study. The response rate was 88.7% for students from the University of Split School of Medicine, 83.1% for students from the University Department of Health Studies, and 65.9% for students from the University of Split Faculty of Maritime Studies.

Students filled a questionnaire that included questions about their age, sex, enrolled faculty, and lifestyle habits (dietary pattern, smoking, sleeping habits, physical activity, and perceived stress), as described in our previous study [41]. Briefly, students were asked to grade their health on a Likert scale from 0 to 10, where 0 represented ill health and 10 represented complete health. Furthermore, students were asked about their self-measured body mass and height, which we used to calculate body mass index (BMI), by dividing body mass (kg) by the square of the value of body height (m^2^). Smoking habits were assessed with one question, and students were divided into 3 subgroups (never smoked, ex-smokers, and active smokers).

In part of the questionnaire that contained questions about dietary habits, we asked participants about their breakfast intake frequency (how many days per week they usually eat breakfast). In order to asses student’s compliance to the MD pattern, we used the Mediterranean Diet Serving Score (MDSS) [42], which has been shown to be a valid instrument and easy to use in the Croatian population [43]. In short, MDSS is measuring the intake of 14 food groups per meal—daily or weekly. Questionnaire emphasizes the importance of the consummation of foods such as fruit, vegetables, olive oil, and cereals, which should be eaten at every main meal each day (3 points are awarded for each of those food groups in case of adherence to recommendations). Daily intake is also required for dairy products and nuts (2 points each), while the remaining groups should be consumed weekly (potato, fish, legumes, eggs, white and red meat, and sweets (1 point is awarded for compliance with each food group). Additionally, adults who drink a glass of wine with one main meal each day receive 1 point. The maximum MDSS score is 24 points, and the proposed cutoff value for MD adherence is 14 points [42]. Details about the MDSS questions and scoring procedure can be found in our previous paper [43]. MDSS questionnaire does not assess the way the food is being prepared (cooking procedures). We also assessed coffee intake, and possible answers were two or more times per day, once a day, three times a week, twice a week, once a week, once a month, or rarely.

Sleeping habits were assessed using a few questions within the self-administered questionnaire, about the usual bedtime, separately for weekdays and weekends, and the usual wake-up time. Based on these questions, we calculated the average sleep duration for weekdays and weekends. This enabled us to calculate the usual time spent sleeping, expressed in hours, as in our previous study [41].

Physical activity was assessed using the International Physical Activity Questionnaire short form (IPAQ; short form) [44]. Based on the students’ responses to this questionnaire, we calculated both the metabolic equivalent of task (MET-min) per week and sitting time during the day (h/day), which was used as a measure of sedentary behavior. In short, the IPAQ short form questionnaire assesses physical activity as either vigorous activity (>6 MET), moderate activity (3–6 MET), or low (<3 MET), as well as it measures sitting. To calculate MET-min values per week, we multiplied the MET value with the number of minutes spent on the activity per day and then by the number of days per week the activity was performed. According to the IPAQ scoring protocol, physical activity levels are categorized into Low Activity Level (fails to meet criteria for moderate and high activity levels), Moderate Activity Level (minimum 600 MET-min/week), and High Activity Level (at least 1500 MET-min/week of vigorous level of activity or 3000 MET-min/week in total).

Perceived stress level was assessed using the Perceived Stress Scale (PSS-10) [45], with possible score ranging between 0 and 40, and higher score is pointing to higher level of stress perceived during the period of one month before participating in the study.

AGEs were measured by skin autofluorescence (SAF) using AGE Reader mu (DiagnOptics Technologies BV, Groningen, The Netherlands). AGE reader is a noninvasive desktop device that uses the fluorescence characteristic of certain AGEs in order to assess the level of accumulated AGEs in the skin [46]. Participant would put the volar side of their dominant forearm on the device, and within 12 s, the result would be obtained in arbitrary units. If a participant had some kind of skin abnormality, the measurement was done on the other forearm. The percentage of autofluorescence was expressed by average intensities of the UV light reflected from the skin [47].

### Statistical Analysis

Categorical variables were presented as absolute numbers and percentages, and numerical variables were presented using median and interquartile range (IQR) due to non-normal data distribution (tested using Kolmogorov–Smirnov test). Chi-square test and Mann–Whitney U test were used in bivariate analysis. Additionally, we created two multivariate linear regression models, with AGEs as outcome (dependent) variable in both models. Predictor variables simultaneously entered in the first model included age, gender, faculty (medical students were considered as a referent group), smoking (active smokers were referent group), BMI, sleep duration during working days and nonworking days, physical activity level (high was a referent group), sitting time, perceived stress, breakfast frequency (number of days per week), coffee intake, and MD adherence. The second regression model included all previously used predictors, except that compliance with all 14 MDSS components was included into the model instead of the overall MD adherence, in order to assess the individual contribution of each food group (olive oil, fruit, vegetables, cereals, nuts, dairy products, legumes, potatoes, eggs, fish, white meat, red meat, sweets, and wine), alongside with coffee intake (entered as ordinal variable). Both models yielded a good model fit, with Durbin–Watson test of 1.882 for the first model (adjusted R^2^ = 38.0%), and 1.974 for the second model (adjusted R^2^ = 38.4%). Additional multivariate linear regression model was created, where self-assessed health perception was the outcome variable, and predictor variables included age, gender, faculty (medical students were considered as a referent group), smoking (active smokers were referent group), BMI, sleep duration, physical activity level (high was a referent group), perceived stress, MD adherence, and AGEs.

Statistical analysis was performed using IBM SPSS Statistics (v21.0; IBM, Armonk, NY, USA), and results with *p* < 0.05 were considered statistically significant.

## 3. Results

### 3.1. Sample Characteristics 

We enrolled a total of 1016 students: 475 were men and 541 were women. The highest proportion of men was recorded among students from the Faculty of Maritime Studies (73.1%), while women were the majority (87.6%) among students at the University Department of Health Studies (Table 1). Men were less commonly never smokers, had a higher average BMI, but also had a better health perception compared to women. The difference between genders was also found in sleeping time during free days, where men slept on average for 8.5 h (IQR = 1.0), and women slept 9.0 h (IQR = 1.5) (*p* < 0.001; Table 1). Men reported higher levels of physical activity, while women reported more sitting time per day. The average perceived stress score in men was lower than in women (median of 17 and IQR = 9.0 vs. 20 and IQR = 10.0; *p* < 0.001). Both men and women reported frequent breakfast consumption, without difference between genders. However, women had a higher average MDSS score compared to men (median of 6.0 and IQR of 6.0 vs. 5.0 and 5.0; *p* < 0.001) and a higher percentage of adherence to the MD (8.3% vs. 3.8%; *p* = 0.003). Women also had a higher average AGEs value when compared to men (median of 1.5 and IQR = 0.3 vs. 1.4 and IQR = 0.3; *p* < 0.001) (Table 1).

Comparison between students who were adherent to the MD and those who were not revealed that adherent students more frequently consumed breakfast (median of 7.0 days and IQR 2.0 vs. median of 6.0 and IQR 3.0; *p* = 0.030) (Table 2). The lowest adherence to the MD was recorded among students from the Faculty of Maritime Studies (3.9%), followed by students from the University Department of Health Studies (7.4%) and medical students (9.9%). The MD adherence was 6.2% in the overall sample. Other investigated lifestyle characteristics did not differ between MD adherent and non-adherent students (Table 2).

### 3.2. Association between the Mediterranean Diet and Advanced Glycation End Products (AGEs)

Multivariate linear regression analysis showed a positive association between AGEs and students’ age (β = 0.547; *p* < 0.001), as well as coffee intake (β = 0.087; *p* = 0.001) (Table 3). We found that female gender was associated with higher AGEs levels compared to men (β = 0.124; *p* < 0.001), which was the same as with maritime students compared to medical students (β = 0.116; *p* = 0.002), and for students with low levels of physical activity compared to the high level (β = 0.060; *p* = 0.033). Lower levels of AGEs were associated with higher breakfast frequency (β = −0.053; *p* = 0.038), and students who never smoked had lower levels of AGEs compared to active smokers (β = −0.146; *p* < 0.001), which was the same with as ex-smokers (β = −0.090; *p* = 0.002). MD adherence expressed as the MDSS score was not associated with AGEs accumulation in the skin among students (β = 0.011; *p* = 0.675) (Table 3).

Additionally, we examined the association between AGEs and the MD components while controlling for important confounding factors (age, gender, faculty, smoking, BMI, sleeping time during working days and free days, physical activity level, sitting time, perceived stress score, breakfast frequency, and coffee intake). Students who were compliant with olive oil intake had higher values of AGEs (β = 0.077; *p* = 0.005; Table 4). Students who were compliant with vegetables intake had lower AGEs (β = −0.061; *p* = 0.038), while the remaining MD food groups did not show a significant association with AGEs levels (Table 4).

Finally, we assessed the association between students’ health perception and lifestyle characteristics and AGEs levels (Table 5). Women rated their health lower than men (β = −0.094; *p* = 0.023), and health studies students rated their health as better than medical students (β = 0.105; *p* = 0.014). Both ex-smokers and those who never smoked rated their health as better compared to active smokers. Students with higher stress levels reported worse health perceptions (β = −0.352; *p* < 0.001). Students who were more compliant with the MD also reported better health perceptions (β = 0.066; *p* = 0.037), while the AGEs levels did not show a significant association with self-rated students’ health (Table 5). Age, BMI, sleep duration, and physical activity level also showed no association with students’ health perception (Table 5).

## 4. Discussion

### 4.1. Diet Components and AGEs Level

This study failed to show an association between AGEs and overall MD compliance in a healthy and young student population from Croatia, who unfortunately demonstrated very low MD compliance. However, we found lower AGEs levels in students who were consuming adequate amounts of vegetables and ate breakfast more regularly, while students who were consuming olive oil adequately and those with higher coffee intake presented with higher levels of AGEs accumulation in the skin.

Several previous studies have shown the beneficial effect of the MD on lowering AGEs levels [18,48,49,50]. One such study identified that a high intake of vegetables, fruits, and nuts and a low intake of sugar-sweetened soft beverages were independently associated with lower AGEs levels assessed via skin autofluorescence [49]. Furthermore, one study showed that adherence to the MD significantly reduced the serum levels of AGEs, which was associated with a higher probability of type 2 diabetes remission [48]. A recent systematic review and metaanalysis of randomized controlled trials showed that a low intake of dietary AGEs was associated with lower insulin resistance, fasting insulin, and a healthier lipid profile [51]. However, it is underscored that dietary patterns low in AGEs might be more important in individuals with cardiometabolic risk factors [51].

However, a recent study in kidney transplant recipients sampled from the same target population as our students, and using the same methodology, also showed no association between the MDSS score and AGEs [52]. One possible reason why we failed to identify an association between the MD and AGEs is very low adherence to the MD among students, which was as low as 3.9% among maritime students, 7.4% among health studies students, and 9.9% among medical students. Unfortunately, this kind of low adherence to the traditional way of eating in the population from southern Croatia is not entirely a new finding [53,54], and it is confirming a systematic departure from a traditional way of life, especially among younger generations [54,55]. This alarming situation should be taken into serious consideration, since it highlights the necessity for urgent interventions in order to preserve the Mediterranean diet and tradition in this area. This is especially important because of lost potential and opportunity for both individual and population health protection, given the benefits of the MD for both physical and mental health [56]. Even in our sample of young and healthy students, we found a positive association between better MD compliance and health perception.

Among MD food components, we found a positive association between adequate olive oil intake and AGEs levels. This is in contrast to our expectations, given that olive oil was shown to have anti-inflammatory and antioxidant properties, especially extra-virgin olive oil [57,58]. However, just because a diet is rich in olive oil does not automatically imply that it is not high in dietary AGEs. As we have shown in this study, our participants did not adhere to the MD; thus, their dietary pattern may be high not only in monounsaturated fatty acids that can be found in olive oil, but it may also be rich in saturated fatty acids, typically present in the Western dietary pattern, which is associated with elevated glycation markers [18]. The MD is considered not to be simply a diet but a dietary pattern, which recognizes that foods are consumed in complex combinations. Thus, an individual component, such as olive oil, may not be enough to result in the beneficial effects of the whole diet. Furthermore, in this study, we did not evaluate the use of different cooking methods. If olive oil consumed by our participants was used for grilling, broiling, frying, and roasting food, that could explain our findings, since these methods are associated with higher levels of dietary AGEs [14]. Additionally, there is a substantial lack of studies specifically quantifying the impact of olive oil on AGEs accumulation in tissues. Hence, further studies are needed in order to elucidate and explain our findings.

Our finding of a negative association between vegetables intake and AGEs levels is in line with previous findings [6,49]. The pigments found in vegetables are antioxidants and bioflavonoids, which have therapeutic effects against several ailments [59]. Polyphenols inhibit glucose absorption, stimulate insulin secretion, reduce hepatic glucose output, and enhance glucose uptake [60]. Additionally, dietary fiber has an effect on the modulation of the postprandial glucose response, and its fermentation by intestinal bacteria results in the production of short chain fatty acids, which improve insulin signaling and sensitivity, as well as glucose response [61]. These mechanisms may be behind the association between higher vegetables intake and lower levels of AGEs, which was found in our study and in several other studies as well [40,49].

We have observed that students who consumed breakfast more frequently had lower levels of AGEs. These results are in line with the findings by Isami et al., who found that eating breakfast every day and the avoidance of sugary foods had an effect on lowering AGEs levels [40]. On the other hand, skipping breakfast was associated with poorer glycemic control and higher levels of HbA1c [62,63]. Skipping breakfast causes higher glucose variability during the day, impaired insulin sensitivity, and increased fat oxidation despite higher postprandial insulin concentrations, which increases metabolic risk over time. Longer fasting periods caused by breakfast skipping increase inflammasome activity and inflammatory responses [64].

### 4.2. Coffee, Smoking, and AGEs Level

We have also found a positive association between coffee consumption and AGEs levels. Several previous studies displayed the same result [34,37,59,65]. Van Waateringe et al. described a positive correlation between coffee consumption and SAF levels, which was dose dependent in both a nondiabetic population and in people with type 2 diabetes [35]. This effect may be due to the fluorescent substances naturally found in coffee [fluorophores] or due to the Millard reaction that happens as a result of coffee bean roasting [37]. AGEs are present in soluble form in beverages, and in this way they are more bioavailable compared to those in solid foods, which could be the reason why coffee has such an effect on AGEs [58]. Additionally, coffee consumers are also more likely to smoke, which could be another reason for higher AGEs levels in coffee consumers [35,60].

Indeed, we detected lower levels of AGEs in subjects who do not smoke and those who are ex-smokers, which is in line with previous findings of cigarette smoke as one of the main sources of exogenous AGEs. A large cross-sectional study on 10 946 volunteers between 20 to 79 years old found the same association between cigarette smoking and AGEs [38]. Van Waateringe et al. found an association between various smoking behaviors and SAF levels, as well as a dose-dependent effect [37]. We have also confirmed their findings that quitting smoking has a reversible effect on SAF levels [37].

The combustion of tobacco products results in the formation of AGEs via the Millard reaction, which produces the key ligands for proinflammatory RAGE signaling [18]. Glycotoxins, highly reactive glycation products that are present in cigarette smoke and aqueous extracts of tobacco, can rapidly induce AGEs formation and cause DNA mutations. These glycotoxines are highly reactive and induce AGEs formation within hours, while glucose and glucose 6-phosphate induce same formation within a period of days to weeks [14].

### 4.3. Physical Activity and AGEs Level

Our results seem to be in the line with the study by Isami et al., who also found a positive association between lower levels of physical activity and AGEs [40]. It was found that higher levels of activity and cardiorespiratory fitness in children had effects on reducing the formation of AGEs [66]. It is assumed that exercise can reverse the formation of early glycation products and induce the formation of soluble RAGE, which then bind to AGEs and act as a competitive inhibitor of ligands that activate RAGE [57]. Physical activity also has an effect on the activation of glutathione reductase, which takes part in protection against oxidative injuries [58].

### 4.4. Age and Gender and AGEs Level

As expected, age was associated with higher levels of AGEs. Aging is perceived as an important factor in the nonenzymatic glocalization of proteins, as confirmed in several previous studies, as well as in ours [36,37,40]. The association between age and AGEs may be due to the reduction of the expression and activation of antioxidative enzymes, such as glyoxalase, which is a crucial enzyme for the detoxification of methylglyoxal [40].

A positive association between female gender and AGEs has also been detected in several previous studies [67,68,69]. Lutgers et al. detected higher autofluorescence in women aged <56 years with type 2 diabetes, which suggests an estrogen-related effect [69]. Sex hormones have an influence on the collagen turnover rate, which could be the reason for the loss of difference in AGEs accumulation between men and postmenopausal women [69]. Gender differences in AGEs accumulation may be also associated with vitamin D deficiency, which is more common in women, while higher concentrations of vitamin D has been associated with lower levels of AGEs [68].

### 4.5. Limitations

This study has some limitations that need to be highlighted. Due to its cross-sectional design, this prevents us from making any causal inferences. Secondly, we used a questionnaire in which we relied on participants’ memory, which could result in recall bias. We did not measure the serum levels of AGEs, as we opted for a noninvasive skin autofluorescence method, which has nevertheless been shown to correlate well with measures of arterial stiffness and atherosclerosis and can be a valuable biomarker for major adverse cardiovascular events [70,71]. Furthermore, the results of this study cannot be generalized, because it included only young and generally healthy students.

The tool used for AGEs assessment in this study was previously validated and has been mainly used for the evaluation of the cardiovascular risk. Its application in nutrition, as in the context of the present study, may need further specific validation, especially for the association with the Mediterranean diet, along with other dietary factors. Hence, this study may prompt further research in this field.

Another issue worth pointing out is that MDSS questionnaire does not assess the cooking procedures, which are very important for the processes of AGEs formation in food. However, the Mediterranean diet and the MDSS questionnaire put high emphasis on the intake of foods with naturally low potential for AGEs formation, such as fruit, vegetables, olive oil, cereals, dairy products, and nuts.

The AGE Reader mu (DiagnOptics) is easy to use, providing quick and reproducible results. Despite the relatively high cost for the initial investment, it offers several clinical applications and benefits: noninvasive measurement, cardiovascular risk assessment, diabetes management, and the early detection of complications. It may be used to motivate patients to adhere to lifestyle changes by demonstrating the impact of diet, exercise, and other interventions on AGEs levels. Further research is required to determine whether the AGE Reader mu is a useful and cost-effective tool for assessing individual dietary quality and for measuring the impact of diet on the levels of AGEs.

The advantages of this study include its relatively high sample size and high response rate, which gives us insight into the dietary habits of students studying in Split, Croatia and shows the extent to which they follow the principles of the Mediterranean diet, which was considered to be a traditional type of diet in this area.

## 5. Conclusions

Our results confirmed some of the previous findings regarding AGEs accumulation and lifestyle characteristics. These results indicate that various lifestyle habits are associated with AGEs accumulation, even in generally healthy and young people. Hence, health promotion and preventive measures are necessary from an early age, especially given the fact that MD compliance was extremely low in our students.

## Figures and Tables

**Table 1 nutrients-16-02483-t001:** Students’ characteristics according to sex (N = 1016).

	Men N = 475	Women N = 541	*p* Value
Age (years); median (IQR)	21.0 (3.0)	21.0 (5.0)	0.673
Faculty; N (%)			<0.001
Medicine	69 (32.5)	143 (67.5)	
Health studies	37 (12.4)	262 (87.6)	
Maritime studies	369 (73.1)	136 (26.9)	
Smoking; N (%)			0.016
Active smokers	155 (50.3)	153 (49.7)	
Ex-smokers	89 (52.7)	80 (47.3)	
Never smoked	221 (42.2)	303 (57.8)	
BMI (kg/m^2^); median (IQR)	24.2 (2.9)	21.6 (3.2)	<0.001
Health perception; median (IQR)	9.0 (2.0)	8.0 (1.0)	<0.001
Sleeping time during working days (h); median (IQR)	7.2 (1.5)	7.3 (1.5)	0.706
Sleeping time during free days (h); median (IQR)	8.5 (1.0)	9.0 (1.5)	<0.001
Physical activity level; N (%)			<0.001
Low	104 (40.0)	156 (60.0)	
Moderate	123 (35.3)	225 (64.7)	
High	248 (60.8)	160 (39.2)	
Sitting time (h/day); median (IQR)	4.5 (4.0)	5.0 (4.5)	0.008
Perceived stress score; median (IQR)	17.0 (9.0)	20.0 (10.0)	<0.001
Breakfast frequency (times/week); median (IQR)	6.0 (3.3)	7.0 (3.0)	0.272
MDSS score; median (IQR)	5.0 (5.0)	6.0 (6.0)	<0.001
MD adherent (MDSS > 14); N (%)	18 (3.8)	45 (8.3)	0.003
AGEs; median (IQR)	1.4 (0.3)	1.5 (0.3)	<0.001

IQR—interquartile range; MDSS—Mediterranean Diet Serving Score; MD—Mediterranean diet; AGEs—advanced glycation end products.

**Table 2 nutrients-16-02483-t002:** Students’ characteristics according to the MD adherence.

	MD Adherent Students (MDSS ≥ 14) N = 63 (6.2%)	MD Non-Adherent Students (MDSS < 14) N = 953 (93.8%)	*p* Value
Age (years); median (IQR)	21.5 (4.0)	21.0 (3.0)	0.673
Faculty; N (%)			<0.001
Medicine	21 (9.9)	191 (90.1)	
Health studies	22 (7.4)	275 (92.6)	
Maritime studies	20 (3.9)	487 (96.1)	
Smoking; N (%)			0.016
Active smokers	14 (4.5)	296 (95.5)	
Ex-smokers	8 (4.7)	161 (95.3)	
Never smoked	41 (7.9)	481 (92.1)	
BMI (kg/m^2^); median (IQR)	22.6 (3.4)	22.9 (3.8)	<0.001
Health perception; median (IQR)	9.0 (1.0)	9.0 (2.0)	<0.001
Sleeping time during working days (h); median (IQR)	7.0 (1.2)	7.3 (1.5)	0.706
Sleeping time during free days (h); median (IQR)	8.0 (1.0)	9.0 (1.5)	<0.001
Physical activity level; N (%)			<0.001
Low	12 (4.6)	247 (95.4)	
Moderate	23 (6.6)	326 (93.4)	
High	28 (6.9)	380 (93.1)	
Sitting time (h/day); median (IQR)	4.0 (4.0)	5.0 (4.5)	0.008
Perceived stress score; median (IQR)	16.5 (11.0)	19.0 (9.0)	<0.001
Breakfast frequency (times/week); median (IQR)	7.0 (2.0)	6.0 (3.0)	0.272
AGEs; median (IQR)	1.5 (0.5)	1.5 (0.4)	<0.001
Age (years); median (IQR)	21.5 (4.0)	21.0 (3.0)	0.003

IQR—interquartile range; MDSS—Mediterranean Diet Serving Score; MD—Mediterranean diet; AGEs—advanced glycation end products.

**Table 3 nutrients-16-02483-t003:** Association between AGEs and lifestyle characteristics using multivariate linear regression analysis.

	Unstandardized Coefficients			Standardized Coefficients	*p* Value
	B	95% CI Lower Bound	95% CI Upper Bound	Beta	
Age	0.038	0.035	0.042	0.547	<0.001
Women (men are referent group)	0.080	0.038	0.121	0.124	<0.001
Faculty (Medical studies is referent group)					
Health studies	0.037	−0.011	0.084	0.052	0.130
Maritime studies	0.075	0.028	0.122	0.116	0.002
Smoking (active smokers are referent group)					
Ex-smokers	−0.077	−0.126	−0.029	−0.090	0.002
Never smoked	−0.095	−0.135	−0.054	−0.146	<0.001
BMI (kg/m^2^)	0.002	−0.003	0.008	0.022	0.424
Sleeping time during working days (h/night)	0.005	−0.009	0.018	0.018	0.508
Sleeping time during free days (h/night)	−0.003	−0.015	0.009	−0.012	0.654
Physical activity level (high is referent group)					
Low	0.044	0.004	0.085	0.060	0.033
Moderate	0.009	−0.029	0.046	0.013	0.652
Sitting time (h/day)	0.003	−0.002	0.008	0.030	0.245
Perceived stress score (PSS-10)	0.001	−0.001	0.004	0.030	0.262
Breakfast frequency (times/week)	−0.008	−0.015	0.000	−0.053	0.038
Coffee intake	0.005	0.002	0.008	0.087	0.001
MD adherence (MDSS points)	0.014	−0.052	0.080	0.011	0.675

MDSS—Mediterranean Diet Serving Score; 95% CI—95% confidence interval.

**Table 4 nutrients-16-02483-t004:** Association between AGEs and adherence to the Mediterranean food groups using multivariate linear regression analysis (model was adjusted for age, gender, faculty, smoking, BMI, sleeping time during working days and free days, physical activity level, sitting time, perceived stress score, breakfast frequency, and coffee intake).

	Unstandardized Coefficients			Standardized Coefficients	*p* Value
	B	95% CI Lower Bound	95% CI Upper Bound	Beta	
Fruit intake	−0.002	−0.017	0.013	−0.007	0.798
Vegetables intake	−0.017	−0.032	−0.001	−0.061	0.038
Cereals intake	−0.009	−0.020	0.003	−0.039	0.131
Olive oil intake	0.025	0.008	0.042	0.077	0.005
Nuts intake	−0.003	−0.027	0.021	−0.008	0.782
Dairy intake	0.000	−0.019	0.019	−0.001	0.971
Potato intake	0.020	−0.030	0.070	0.020	0.433
Legumes intake	−0.005	−0.040	0.030	−0.008	0.774
Eggs intake	0.004	−0.029	0.036	0.006	0.818
Fish intake	0.018	−0.021	0.056	0.025	0.361
White meat intake	−0.004	−0.045	0.037	−0.005	0.838
Red meat intake	0.010	−0.029	0.050	0.014	0.611
Sweets intake	0.011	−0.032	0.055	0.015	0.608
Wine intake	0.019	−0.055	0.092	0.013	0.616

95% CI—95% confidence interval.

**Table 5 nutrients-16-02483-t005:** Association between health perception and lifestyle characteristics and AGEs using multivariate linear regression analysis.

	Unstandardized Coefficients			Standardized Coefficients	*p* Value
	B	95% CI Lower Bound	95% CI Upper Bound	Beta	
Age	−0.018	−0.042	0.006	−0.056	0.151
Women (men are referent group)	−0.276	−0.513	−0.038	−0.094	0.023
Faculty (Medical studies is referent group)					
Health studies	0.334	0.069	0.599	0.105	0.014
Maritime studies	0.248	−0.015	0.511	0.085	0.065
Smoking (active smokers are referent group)					
Ex-smokers	0.286	0.017	0.554	0.074	0.037
Never smoked	0.325	0.110	0.540	0.111	0.003
BMI (kg/m^2^)	−0.019	−0.051	0.013	−0.040	0.242
Sleeping time during working days (h/night)	0.058	−0.019	0.135	0.050	0.140
Sleeping time during free days (h/night)	0.039	−0.031	0.110	0.037	0.275
Physical activity level (high is referent group)					
Low	0.015	−0.217	0.247	0.004	0.900
Moderate	−0.128	−0.340	0.084	−0.042	0.237
Perceived stress score (PSS-10)	−0.073	−0.087	−0.060	−0.352	<0.001
MD adherence (MDSS points)	0.026	0.002	0.050	0.066	0.037
AGEs	−0.045	−0.387	0.297	−0.010	0.796

BMI—Body mass index, MDSS—Mediterranean Diet Serving Score, AGEs—Advanced glycation end products, 95%CI—95% confidence interval.

## Data Availability

The data presented in this study are available upon request from the corresponding author due to privacy reasons.

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
