# Peer review of "Association between Mediterranean Diet and Advanced Glycation End Products in University Students: A Cross-Sectional Study"

_nutrients, 2024, doi:10.3390/nu16152483_

Round 1

Reviewer 1 Report

Comments and Suggestions for Authors

The paper is well set up and addresses an interesting and relevant topic, but could be further improved.

Usefulness of the instrument for assessing AGEs: The paper uses the AGE Reader mu (DiagnOptics) to measure the accumulation of AGEs in the skin through autofluorescence. This technique has been validated in several previous studies and is useful for the non-invasive assessment of AGEs in tissue. However, it is important to note that this tool is often mainly used in the areas of cardiovascular risk (cdv) and chronic diseases. Its application in nutrition, as in the context of the present study, may need further specific validation for association with the Mediterranean diet and other dietary factors.

Evidence of use in nutrition: The paper cites a few studies showing the beneficial effect of the Mediterranean diet in reducing AGEs levels. However, most previous research focuses on the effects of AGEs in cardiovascular and metabolic risk contexts. The use of the tool in a nutritional study context could be innovative, but it is necessary to emphasise the lack of extensive specific evidence supporting the use of the AGE Reader mu exclusively to assess the effect of specific diets such as the Mediterranean diet.

Production of AGEs through diet: AGEs are formed through the Maillard reaction, where carbonyl components of sugars react with free amino groups of amino acids. Foods at high risk for AGEs formation include those rich in fat and protein, cooked at high temperatures or for long periods, such as fried, grilled, baked foods and animal products.

Completeness of the AGEs food survey: The survey used should be carefully checked to ensure that it includes all categories of foods that contribute most to AGEs, especially those produced by inappropriate cooking methods. The methodology of the questionnaire should be detailed to ensure representativeness and accuracy in identifying sources of AGEs in students' diets.

Instrument costs and clinical utility: The discussion of the paper could benefit from the inclusion of a paragraph on the costs associated with the use of the AGE Reader mu and its possible utility in the clinical setting. This could include a cost-effectiveness analysis and the potential of the AGE Reader mu as a screening tool for AGEs-related disease risk, thereby improving preventive and therapeutic management in the clinic. Translated with www.DeepL.com/Translator (free version)

Comments on the Quality of English Language

English is fine

Author Response

Dear sir/madam,

We would like to thank you for your comments and the opportunity to further improve our manuscript, as well as for detailed review of our manuscript. Your comments are highly appreciated and have been helpful to us. Please find our responses below.

  1. Usefulness of the instrument for assessing AGEs: The paper uses the AGE Reader mu (DiagnOptics) to measure the accumulation of AGEs in the skin through autofluorescence. This technique has been validated in several previous studies and is useful for the non-invasive assessment of AGEs in tissue. However, it is important to note that this tool is often mainly used in the areas of cardiovascular risk (cdv) and chronic diseases. Its application in nutrition, as in the context of the present study, may need further specific validation for association with the Mediterranean diet and other dietary factors.

Response: We agree that the AGEs reader has not been previously used in nutrition science to measure the impact of diet on the levels of advanced glycation end-products, and that it should be validated for this application. We hope that our study will prompt new research focused specifically on this topic. We are confident that this could become a useful and cost-effective tool for assessing individual dietary quality. We have added this text in the Introduction, in order to highlight your comment, with which we agree:

Additionally, a non-invasive tool for the assessment of AGEs in tissue has been frequently used (AGE Reader mu; DiagnOptics), and it has been validated and shown to be useful in the area of cardiovascular risk assessment. However, this device has not been extensively used in nutrition studies for association between AGEs and the Mediterranean diet or other dietary factors. Hence, there is a knowledge gap among younger individuals, who are still free from chronic diseases, especially regarding the relevance of AGEs for their health, as well as the interplay of different factors associated with AGEs levels, such as nutrition and other lifestyle habits. The aim of this study was to investigate the association between AGEs accumulation in the skin and the MD pattern, as well as other lifestyle characteristics in the population of young and healthy university students from Croatia.” (lines 113-124)

  1. Evidence of use in nutrition: The paper cites a few studies showing the beneficial effect of the Mediterranean diet in reducing AGEs levels. However, most previous research focuses on the effects of AGEs in cardiovascular and metabolic risk contexts. The use of the tool in a nutritional study context could be innovative, but it is necessary to emphasise the lack of extensive specific evidence supporting the use of the AGE Reader mu exclusively to assess the effect of specific diets such as the Mediterranean diet.

Response: Thank you for this comment. We have added this text in the Limitations section, in order to highlight your comment:

“The tool used for AGEs assessment in this study was previously validated and mainly used for evaluation of the cardiovascular risk. Its application in nutrition, as in the context of the present study, may need further specific validation, especially for the association with the Mediterranean diet, and other dietary factors. Hence, this study may prompt further research in this field.” (lines 435-439)

  1. Production of AGEs through diet: AGEs are formed through the Maillard reaction, where carbonyl components of sugars react with free amino groups of amino acids. Foods at high risk for AGEs formation include those rich in fat and protein, cooked at high temperatures or for long periods, such as fried, grilled, baked foods and animal products.

Completeness of the AGEs food survey: The survey used should be carefully checked to ensure that it includes all categories of foods that contribute most to AGEs, especially those produced by inappropriate cooking methods. The methodology of the questionnaire should be detailed to ensure representativeness and accuracy in identifying sources of AGEs in students' diets.

Response: Thank you for pointing this out. However, this particular questionnaire is quite short, and it only investigates the frequency of intake of certain food groups that are necessary for inclusion in the Mediterranean diet. We have added this information in the Methods section:

„MDSS questionnaire does not asses the way the food is being prepared (cooking procedures).“ (lines 167-168)

We have also included this into Limitations section:

“Another issue worth pointing out is that MDSS questionnaire does not asses the cooking procedures, which are very important for the processes of AGEs formation in the food. However, Mediterranean diet and the MDSS questionnaire put high emphasis on the intake of foods with naturally low potential for AGEs formation, such as fruit, vegetables, olive oil, cereals, dairy products and nuts.” (lines 440-444)

  1. Instrument costs and clinical utility: The discussion of the paper could benefit from the inclusion of a paragraph on the costs associated with the use of the AGE Reader mu and its possible utility in the clinical setting. This could include a cost-effectiveness analysis and the potential of the AGE Reader mu as a screening tool for AGEs-related disease risk, thereby improving preventive and therapeutic management in the clinic. Translated with www.DeepL.com/Translator (free version)

Response: Thank you for this comment. We have added this text in the Limitations section, in order to highlight your comment:

AGE Reader mu mu (DiagnOptics) is easy to use, providing quick and reproducible results. Despite the relatively high cost for the initial investment, it offers several clinical applications and benefits: non-invasive measurement, cardiovascular risk assessment, diabetes management and early detection of complications. It may be used to motivate patients to adhere to lifestyle changes by demonstrating the impact of diet, exercise, and other interventions on AGEs levels. Further research is required to determine whether AGEs Reader mu is a useful and cost-effective tool for assessing individual dietary quality, and measuring the impact of diet on the levels of AGEs. (lines 445-452)

Reviewer 2 Report

Comments and Suggestions for Authors

I have several concerns on this manuscript, as follows:

1. Introduction: "Besides diet and physical activity, previous studies have identified the effects of several 104 other factors on the level of AGEs. These include the positive association between AGEs 105 and age, coffee consumption, [36-38], and cigarette smoking [37-39]. Smoking has a sig-106 nificant effect on accumulation of AGEs, which is affected by the pack-years and the num-107 ber of hours being exposed to second-hand smoking, indicating a dose-dependent effect 108 [37][39]. It has also been found that if a person quits smoking, the AGEs level will gradu-109 ally decrease over time" Why did the authors add this paragraph? Please clarify it.

2. Methods: For participants, how did the authors recruit them? and How did the participants know this study? Please added more information about these.

3. For Sleeping habits, how did the authors assess it? by interview? or by questionnaire? Please add more information.

4. For physical activity, how did the authors calculate the MET-min? Please add more explanation.

5. Abbreviations in the tables need to be explained in full words below the tables.

6. For discussion part, the authors should re-organize the structure of discussion content.

Author Response

Dear sir/madam,

We would like to thank you for your comments and the opportunity to further improve our manuscript, as well as for detailed review of our manuscript. Your comments are highly appreciated and have been helpful to us. Please find our responses below.

I have several concerns on this manuscript, as follows:

  1. Introduction: "Besides diet and physical activity, previous studies have identified the effects of several 104 other factors on the level of AGEs. These include the positive association between AGEs 105 and age, coffee consumption, [36-38], and cigarette smoking [37-39]. Smoking has a sig-106 nificant effect on accumulation of AGEs, which is affected by the pack-years and the num-107 ber of hours being exposed to second-hand smoking, indicating a dose-dependent effect 108 [37][39]. It has also been found that if a person quits smoking, the AGEs level will gradu-109 ally decrease over time" Why did the authors add this paragraph? Please clarify it.

Response: Thank you for your comment. The mentioned paragraph was included to emphasize the impact of other lifestyle factors that are relevant for AGEs accumulation, some of which we have also included in our results. Hence, we provided a short introduction for these risk factors. For instance, smoking effect on AGEs was shown to be dose-dependent. Our study also demonstrated an association between the levels of advanced glycation end-products, with participants categorized based on their smoking status into current smokers, former smokers, and never smokers, which is in line with cited studies (references 39 and 40).

  1. Methods: For participants, how did the authors recruit them? and How did the participants know this study? Please added more information about these.

Response: Thank you for your question. We have added more details in this section:

“The researchers enrolled students during their regular and obligatory courses. Firstly, we obtained consent from both the professors and the students for participation, informing them about the procedures and the purpose of the study.“ (lines 134-136)

  1. For Sleeping habits, how did the authors assess it? by interview? or by questionnaire? Please add more information.

Response: Sleep habits were assessed using a questionnaire. We did not use a structured questionnaire; instead, data were collected through a few simple questions about the usual bedtime, separately for weekdays and weekends, and the usual wake-up time. Based on these questions, we calculated the average sleep duration for weekdays and weekends.

We have added these details in the Methodology section, lines 172-175:

"Sleeping habits were assessed using a few questions within the self-administered questionnaire, about the usual bedtime, separately for weekdays and weekends, and the usual wake-up time. Based on these questions, we calculated the average sleep duration for weekdays and weekends.“

  1. For physical activity, how did the authors calculate the MET-min? Please add more explanation.

Response: Thank you for your question. Within the questionnaire we used the International Physical Activity Questionnaire (IPAQ), short form, from which we calculated the Metabolic equivalent of task (MET-min). We have added this text:

“Physical activity was assessed using the International Physical Activity Questionnaire, short form (IPAQ; short form) [44]. Based on the students’ responses to this questionnaire, we calculated both the metabolic equivalent of task (MET-min) per week, and sitting time during the day (h/day), which was used as a measure of sedentary behavior. In short, the IPAQ short form questionnaire assesses physical activity as either vigorous activity (>6 MET), moderate activity (3-6 MET), or low (<3 MET), and sitting. To calculate MET-minutes per week, we multiplied the MET value with the number of minutes spent on the activity per day and then by the number of days per week the activity was performed. According to the IPAQ scoring protocol, physical activity levels are categorized into: Low Activity Level (fails to meet criteria for moderate and high activity levels), Moderate Activity Level (minimum 600 MET-min/week), High Activity level (at least 1500 MET-minutes/week of vigorous level of activity or 3000 MET-minutes/week in total).” (lines 179-191)

  1. Abbreviations in the tables need to be explained in full words below the tables.

Response: Thank you for your observations. We have added the necessary corrections below the tables.

  1. For discussion part, the authors should re-organize the structure of discussion content.

Response: We have added sub-sections into the Discussion section, in order to make it clearer. We have also added more information under the Limitations section.

Round 2

Reviewer 1 Report

Comments and Suggestions for Authors

They improved the paper following my instructions.

I suggest that in future work, the firing methods used should be evaluated. It is a key element.